# Malignant Neoplasms Arising in the Cardiac Pacemaker Cavity: A Systematic Review

**DOI:** 10.3390/cancers15215206

**Published:** 2023-10-29

**Authors:** Francisco Cezar Aquino de Moraes, Lucca Dal Moro, Fernando Rocha Pessoa, Ellen Sabrinna dos Remédios Passos, Raul Antônio Lopes Silva Campos, Dilma do Socorro Moraes de Souza, Danielle Feio, Rommel Mario Rodríguez Burbano, Marianne Rodrigues Fernandes, Ney Pereira Carneiro dos Santos

**Affiliations:** 1Oncology Research Center, University Hospital João de Barros Barreto, Belém 66073-005, PA, Brazil; dra.daniellefeio@gmail.com (D.F.); fernandesmr@yahoo.com.br (M.R.F.); npcsantos.ufpa@gmail.com (N.P.C.d.S.); 2Department of Medicine, Federal University of Pará, Belém 66075-110, PA, Brazil; luccadalmoro98765@gmail.com (L.D.M.); fernandopessoarocha@gmail.com (F.R.P.); ellensasa09@gmail.com (E.S.d.R.P.); ls.campos01@gmail.com (R.A.L.S.C.); dsouza@cardiol.br (D.d.S.M.d.S.); 3Gaspar Vianna State Public Hospital of Clinical Foundation, Belém 66083-106, PA, Brazil; 4Otávio Lobo Children’s Cancer Hospital, Belém, 66063-005, PA, Brazil; rommelburbano@gmail.com

**Keywords:** artificial pacemaker, neoplasms, malignancies, pacemaker complication

## Abstract

**Simple Summary:**

Cancer, a complex group of diseases marked by abnormal cell proliferation and loss of physiological functions. Furthermore, tumors in the cardiac pacemaker pocket are rare and challenging medical problems, where the location of the implanted devices employed to manage heart rhythm disorders unexpectedly becomes a site for neoplasm growth. This intersection becomes even more nuanced with the rise in cardiac pacemaker (PM) implantations, a common practice globally. This study aimed to evaluate reported cases of this condition throughout the existing literature, in addition to enhancing early detection strategies and improving the management of affected patients.

**Abstract:**

Cancer is the abnormal proliferation of physiologically inadequate cells. Studies have identified the cardiac pacemaker pocket as a site of rare neoplasms. To evaluate the clinical outcomes, treatment, prognosis, and individualized management of tumors originating in the cardiac pacemaker pocket, a systematic review was conducted using case reports and case series available in the PubMed/Medline, Science Direct, Cochrane Central, LILACS, and Scientific Electronic Library Online (Scielo) databases. Pacemaker pocket tumors affected patients with a mean age of 72.9 years, with a higher incidence in males (76.9%, n = 10). The average time for neoplasm development was 4.4 years (54.07 months). The most prevalent model was Medtronic (38.4%, n = 5), with titanium (83.3%) being the most common metal composition. Chemotherapy was the most performed procedure among patients (38.4%), followed by radiation therapy (38.4%) and surgical tumor resection (30.7%). Six analyzed cases (46.1%) resulted in death, and four patients (30.7%) achieved a cure. Patients with pacemakers should be routinely evaluated for the occurrence of malignant tumors at the site of device implantation.

## 1. Introduction

Cancer is a cluster of diseases characterized by the abnormal proliferation of cells, leading to the loss of their physiological functions in terms of division, growth, and lifespan [1,2]. Oncogenesis involves intricate mechanisms influenced by genetic factors, environmental exposures, and individual lifestyle habits that contribute to the development of malignancy [3,4]. In the United States, an estimated 2 million new cancer cases are diagnosed annually, with 610,000 deaths attributed to the disease [5]. Globally, approximately 19.3 million cases are identified each year, accompanied by 10 million deaths [6].

Cardiovascular diseases (CVDs) and cancer rank as the primary causes of global mortality [7], accounting for approximately one in six deaths worldwide [8]. The correlation between anticancer therapies and alterations in left ventricular ejection function, as well as the onset of heart failure symptoms, is well documented [9]. Factors such as radiotherapy and certain medications (anthracyclines, cyclophosphamide, sunitinib) [10] have been shown to induce senescence in cardiomyocytes. Consequently, this association is justified by the exacerbation of adverse cardiac remodeling due to the secretion of pro-inflammatory molecules and matrix protease degradation, which significantly impact patient prognosis [11,12].

The cardiac pacemaker (PM), on the other hand, is an electronic device utilized to regulate heart rhythm and treat electrical conduction disorders, including bradycardia, atrioventricular blocks (AVBs), left bundle branch blocks (LBBBs), right bundle branch blocks (RBBBs), and other congenital or acquired cardiovascular diseases (CVDs) [13,14]. Over time, the use of pacemakers has progressively increased, and their implantation is associated with a reduced risk of cardiovascular complications such as heart failure, acute myocardial infarction, malignant arrhythmias, and even mortality [15,16].

Pacemaker implantation is progressively increasing. More than 1 million cardiac pacemakers are implanted per year worldwide, with 200,000 implantations being performed in the United States [17]. Although pacemaker implantation is safe, complications such as bleeding, infection, pain, and inflammation at the incision site are present in a reduced proportion of insertion procedures [18]. In rare cases, the development of malignant neoplasms around the pacemaker tissue adjacent to the pacemaker has also been described [19].

Currently, investigations regarding malignancies in the pacemaker pocket (PP) are restricted to case reports and series. Hence, this systematic review endeavored to assess, by examining reports and case series, the available evidence concerning clinical outcomes, treatment approaches, prognosis, and tailored management strategies for each case.

## 2. Materials and Methods

### 2.1. Protocol and Registration

This systematic review followed the 27 items described in the Preferred Reporting Items for Systematic Review and Meta-Analyses (PRISMA) protocol [20], which assist in the construction of systematic reviews and meta-analyses. This review was registered in the Prospective International Registry of Systematic Reviews (PROSPERO) of the National Institute for Health Research under number CRD42022360240.

### 2.2. Eligibility Criterion

The articles included in this review were only case reports and case series in patients over 18 years of age who registered malignant neoplasms with the primary site of the pacemaker pocket. Only articles with confirmation of cancer by histopathology and/or immunohistochemistry were included in this study. Only articles in English were included in this review.

Articles from literature reviews and encyclopedias, editorials, book chapters, conference abstracts, correspondence, reviews, news, and small communications were excluded. Interventional studies involving animals or humans, and other studies that require ethical approval, must list the authority that provided approval and the corresponding ethical approval code.

Thus, we sought to answer the following question: what are the main clinical, demographic, and management-related characteristics described in cases of malignancy affecting the region covering the pacemaker implantation cavity?

### 2.3. Search Strategy and Data Extraction

This study used references describing cancer at the primary site in the pacemaker pocket indexed in the PubMed/Medline, SCOPUS, Web of Science, and LILACS databases. The following descriptors were used: “Cancer”, “malignancy”, “tumor”, “malignancies”, “carcinoma”, “Plasmacytoma”, “neoplasms”, “neoplasia”, “lymphoma”, “adenocarcinoma”, “leiomyosarcoma”, “histiocytoma”, “artificial pacemaker”, “Resynchronization therapy”, “CIED”, “cardiac implantable electronic devices”, “cardiac pacemaker”, “cardiac pacing artificial”, “implantable pacemaker”, “pacemaker pocket”, “pacemaker cavity”, “pacemaker implantation site”, “pacemaker sac”. For the combination of terms in the databases, we used Boolean connectors (OR, AND). For the inclusion of additional articles, a manual search was performed in the references of the selected studies and notification alerts in the databases were activated if new titles that suited the query were published. In addition, research was also carried out using abstracts, articles, and scientific presentations from virtual meetings of the American Society of Clinical Oncology [21] and the American College of Cardiology [22].

Sources found in the databases and in the references of the articles were incorporated into the reference management software (EndNote^®^, version X7, Thomson Reuters, Philadelphia, PA, USA). Duplicate articles were removed using both automated and manual methods. Subsequently, two reviewers (F.C.A.M. and F.R.P.) independently analyzed the titles and abstracts of the identified articles. In case of disagreements between the two reviewers, a third reviewer was responsible for the final decision (N.P.C.S.).

The following baseline characteristics were extracted: (1) age; (2) sex; (3) pacemaker model; (4) pacemaker composition metal; (5) type of cancer; (6) reported clinical symptoms; (7) comorbidities; (8) period of implementation until the development of cancer; (9) examinations, included laboratory, echocardiogram, electrocardiogram, computerized tomography, mammography, ultrasound, radiography, and histologic; (10) conduct.

### 2.4. Risk of Bias in Included Studies

To assess the risk of bias in the selected articles, the Critical Assessment of the Joanna Briggs Institute (JBI) [23] for case reports was used as a tool, which consists of a checklist of eight scoring items. The evaluation was carried out by two reviewers independently (L.D.M. and R.A.L.S.C.). And in case of disagreement, a third reviewer was responsible for the final opinion (M.R.F.). Additionally, to reduce the risk of bias, all studies included in this review were published in peer-reviewed journals.

Data were tabulated in Microsoft Office Excel version 2016, and patient characteristics, tumor classification, pacemaker implantation time, clinical symptoms, laboratory and imaging tests, management, and outcome were presented descriptively.

## 3. Results

### 3.1. Selection of Studies

We identified 1277 titles of which, after removing duplicates, 819 titles remained for analysis. By applying the eligibility criteria, we selected 22 articles to compose the literature review. As seen on the Figure 1. 

### 3.2. Study Features

Of the 22 selected studies, 6 were carried out in the United States of America (USA): Hamaker et al. [24]; Herrmann, Mishra, and Greenway [25]; Magilligan and Isshak [26]; Moseley et al. [19]; Reyes [27]; and Zarifi et al. [28]. Fourteen were carried out in Europe: 1 in Portugal [29], 2 in Spain [30,31], 2 in the United Kingdom [32,33], 1 in Norway [34], 2 in Italy [35,36], 1 in Germany [37], 1 in Greece [38], 1 in Austria [39], 1 in Slovakia [40], 1 in the Netherlands [41], and 1 in Switzerland [42]. On the Asian continent, two studies were carried out in Japan [43] and one in Israel [44]. The graphic representation of the origin of the case reports can be seen in Figure 2.

### 3.3. Risk of Bias in Studies

In comparing the case reports, 22 articles were determined to be at low risk of bias [24,25,26,27,28,29,30,31,32,33,34,35,37,38,39,40,41,42,43,44]. Only one study was identified as having a high risk of bias [36], as shown in Figure 2 and Figure 3. Respective graphics representations are in Appendix A.

### 3.4. Results of Individual Studies

Eighteen (n = 18) case reports and four (n = 4) case series were included in the systematic review [24,25,26,27,28,29,30,31,32,33,34,35,36,37,38,39,40,41,42,43,44]. Milner et al. [29] reported an ulcerative and expansive plasmacytic lymphoma in the pacemaker pocket (PP) of a 78-year-old man from Portugal. De Mattia, Brieda, and Dametto [36] presented a case of an 87-year-old female patient with an invasive ductal carcinoma developing in the pacemaker pocket, which, along with Moruzzo et al.’s study [35], represented non-Hodgkin’s lymphomas identified consecutively in the adjacent region of the pocket; these were the two reported Italian cases. In the UK, cases were reported by Bhandarkar, Bewu, and Taylor and Rathinam et al. [32,33], describing two adenocarcinomas and one inflammatory myofibroblastic tumor, respectively. The study from Spain conducted by González-Vela et al. in 2009 [32] described a cutaneous leiomyosarcoma in the subpectoral pouch of a 74-year-old man, and another case report published in 2013 [31] described an atypical fibroxanthoma in the PP of an 89-year-old man with four years of implantation. Rasmussen et al. [34] reported a case of papillary adenocarcinoma that developed one year after implantation in a 75-year-old man in Norway. Hamaker et al. [24] reported a 48-year-old male patient with a plasmacytoma in the PP region 16 months after implantation. The study conducted in Germany by Fraedrich et al. [37] described a malignant fibrous histiocytoma in the PP of an 82-year-old male patient three years after implantation. Hojo et al. [43] described, in their Japanese study, a case of a 29-year-old man with stage II diffuse large B-cell lymphoma that developed six years after implantation. The Slovak study by Zonca et al. [40] reported a case of invasive ductal carcinoma in the PP of a 78-year-old woman with ulcerations in the affected region. The Dutch case report by Khamooshian et al. [41] described a pleomorphic sarcoma in the PP of a 43-year-old man eight months into the third device replacement.

In the United States, six cases have been reported. In Hamaker et al. [24], a man was described who developed plasmocytoma, diagnosed 1 year and 4 months after implantation of the generator. Reyes [27] reported in his study a case of clear cell hidradenocarcinoma affecting the pacemaker region of an 88-year-old woman four years later. In the work by Herrmann, Mishra, and Greenway [25], the case of a nodular basal cell carcinoma with features of an erythematous plaque on the left pectoral under the generator was cited. Magilligan and Ishak [26] described in their study a case of an 89-year-old woman who developed a breast adenocarcinoma in the region located in the PP. Zarifi et al. [28] described a case of plasmablastic lymphoma affecting the PP of a 100-year-old male patient after a period of 10 years from implantation until the onset of symptoms, as observed in Table 1 below.

### 3.5. Demographic Characteristics

Years of publication were listed between 1974 and 2021 (Table 1). The country with the most publications was the United States [19,24,25,26,27,28] (n = 6), followed by Spain (n = 2) [30,31], the UK [32,35] (n = 2), Italy (n = 2) [35,37], Austria (n = 1) [39], Germany (n = 1) [37], Slovakia (n = 1) [40], Greece (n = 1) [38], Israel (n = 1) [44], Japan (n = 1) [43], Norway (n = 1) [34], the Netherlands (n = 1) [41], Portugal (n = 1) [29], and Switzerland (n = 1) [42]. The mean age range of patients ranged from 29 to 100 years old. In total, 26 patients were analyzed, 14 men (53.8%) [19,24,25,28,29,30,31,32,35,37,39,41,43] and 12 women [19,24,26,33,36,38,40,42,44] (46.1%).

### 3.6. Malignancies

When analyzing the malignancies, there was a predominance of adenocarcinoma in 29.62% (n = 8) [26,33,34,38,42,44], lymphoma in 22.22% (n = 6) [19,28,29,35,43], and carcinoma in 22.22% (n = 6) [19,25,36,39,40,44]. Adenocarcinomas, were classified as the papillary type (n = 1) [34], breast (n = 1) [26], clear cell hidradenocarcinoma (n = 1) [27], intraductal with extracellular mucus (n = 1) [42], unspecified adenocarcinoma (n = 3), [33,44], and ecchymosal adenocarcinoma (n = 2) [38]. The lymphomas were all of the non-Hodgkin type: lymphoplasmacytic lymphoma (n = 1) [29], stage I–E non-Hodgkin lymphoma (n = 1) [35], stage II diffuse large B-cell lymphoma (n = 1) [43], plasmablastic lymphoma (n = 1) [28], large B-cell lymphoma (n = 1) [19], and B-cell lymphoma (n = 1) [19].

As for carcinomas, they were subdivided into moderately differentiated squamous cell carcinoma (n = 1) [19], invasive ductal carcinoma (n = 3) [36,39,40], nodular basal cell carcinoma (n = 1) [25], and intraductal carcinoma (n = 1) [44]. Other tumors affecting the pacemaker pouch described were: atypical fibroxanthoma [31] (n = 1), plasmacytoma (n = 1) [24], malignant fibrous histioma (n = 1) [37], clear cell hidradenocarcinoma (n = 1) [27], undifferentiated pleomorphic sarcoma (n = 1) [41], cutaneous leiomyosarcoma (n = 1) [30], and inflammatory myofibroblastic tumor (n = 1) [32].

### 3.7. Pacemaker Features

Of the 26 patients, 8 (30.7%) registered the insertion of the Medtronic model (Table 2), with varying subtypes: KVDD 901 [31], 5841 [24], Xytron [37], 5950 [26], Adapt ADDR01 [28], Capsure SP 4024 [39], and 5942 [44]. The most frequent composition metal was titanium in 30.7% (n = 8) [24,26,28,30,31,43,44], followed by mercury zinc with one patient (3.8%), who used the Unipolar Cordis Stanicor model [34] according to the information obtained on the constitution of pacemakers.

### 3.8. Reported Clinical Symptoms

In general, the most frequent clinical manifestation observed was local expansion over or close to the pacemaker pocket (Table 2); 5 patients (19.23%) were registered with local expansion [19,24,28,29,37] and 11 (42.30%) had a palpable mass [26,27,30,31,33,34,35,38,41,42,43,44]. In addition, other symptoms were recorded, such as ulceration in five patients (19.23%) [29,30,31,34,40], necrosis in two (7.7%) [24,43], fever in three (11.53%) [24,32,35], secretion in one (3.8%) [34], infection in one (3.8%) [34], and the presence of erythema in one (3.8%) [19].

### 3.9. Comorbidities

Thirteen patients (50.0%) had a history of AVB [19,24,29,31,34,37,38,41,43,44] and five of systemic arterial hypertension (SAH) (19.2%) [19,28,29,35]. There were five cases of Stokes-Adams syndrome (19.2%) [34,37,38,43], four cases of type 2 diabetes mellitus (DM2) (15.4%) [19,28,29], three cases of atrial fibrillation (AF) (11.5%) [25,29,34], and two cases with a previous cancer (7.7%) [29,44]. These comorbidities are listed in Table 2.

### 3.10. Time Elapsed from Implantation to Development of Cancer

The average time for the evolution of neoplasia was 6.57 years (78.92 months). A total of 12 patients developed cancer between 0 and 48 months after pacemaker implantation [19,24,27,30,31,34,35,37,38,41,44], 5 patients developed some neoplasm between 49 and 96 months [26,36,38,40,43], and 8 had a period longer than 97 months [19,28,29,32,33,39,42]. The graphical representation of these data can be seen below in Figure 4.

### 3.11. Laboratory Tests

There was no predominant alteration among the evaluated cases, as seen in Appendix A. However, the occurrence of reduced hemoglobin [24,29,32], positive QuantiFERON-TB Gold [29]; elevated LDH and lambda IgA [35], increased soluble interleukin 2 (IL-2) receptor [43], elevated CRP [32], increased ESR [28,32], and leukocytosis and changes in biochemical markers of cardiovascular damage [28] should be noted, in addition to leukopenia associated with thrombocytopenia [24].

### 3.12. Cardiovascular Assessment

#### 3.12.1. Echocardiogram

Three patients (11.54%) [29,32,41] underwent echocardiography (ECHO). The results were: one ECHO case indicating mildly reduced ejection fraction [29], Three ECHO case with reduced ejection fraction [41], and one case showing no evidence of endocarditis [32]

#### 3.12.2. Electrocardiogram

Five patients (19.23%) [24,25,26,43,44] recorded an ECG after admission. In all of them, the presence of AVB was observed.

### 3.13. Imaging Exams

In total, 6 cases (23%) underwent CT (Appendix A). Among them, one case had a mass measuring 65 × 24 mm associated with left axillary lymphadenopathy [29]; one case had a mass on the pacemaker battery measuring 6 cm in diameter [35], also with localized lymphadenopathy; and one case had a round mass measuring 8–9 cm in the left clavicular region superficial to the left pectoral muscle on the upper surface of the pacemaker [32]. Furthermore, two suspected metastases were ruled out [41,42] and one was identified as highly suggestive. [38]

Regarding mammography, it was performed in three cases, presenting the following results: spiculated round mass caudal to the pacemaker pocket [36], absence of abnormalities or suspicious changes in both breasts [27], and findings difficult to interpret [40]. Four cases (15.3%) underwent USG, revealing a solid-looking lesion in two cases [27,41] and liquid in one case [28]

A total of seven case reports (26.9%) presented in their descriptions the results of chest X-rays performed [24,26,29,33,37,42], of which two reported no visible changes [26,29], one reported the presence of soft tissue mass around the pulse generator [24], one reported a shadow in the posterobasal segment of the lung with a suspicious indication of bronchial carcinoma with enlargement [37], one reported no metastasis [42], one reported a clinically palpable mass [33], and one reported nodular bilateral round shadows typical of metastatic lesions in the lungs [38].

### 3.14. Histopathological

Histopathological analysis (Appendix A), with the aid of immunohistochemistry, revealed the identification of proteins among the lymphomas (*n* = 6) [19,28,29,43], which acted as markers, the majority (*n* = 3) being positive for CD138 [28,29,35] and for Ki-67 [28,29] (*n* = 2), in addition to other markers, such as CD4 [29], CD30, and CD43 [35]. Expression of CD10, CD99, CD68 (focal) and smooth muscle actin (focal), S100 protein, melan-A, desmin, CD34, p63, CD31, and human herpesvirus latent nuclear antigen 8 were markers also identified in atypical fibroxanthoma [31], together with the presence of hyperchromatic cells described both in atypical fibroxanthoma [31] and in undifferentiated pleomorphic sarcoma [41].

In addition to lymphomas, two other groups were identified in a large portion of the samples; namely, carcinomas (*n* = 6) [19,25,36,39,40,44] of the intraductal (*n* = 2) [39,44], nodular basal [25], ductal (*n* = 3) [36,39,40], and squamous cell types [19] and adenocarcinomas, which accounted for 30.7% of the total (*n* = 8) [26,33,34,38,42], with samples of being not specified (*n* = 4) [28,33,45] and of the schirrous (*n* = 2) [38], intraductal [42], and papillary types [34].

Diagnostic patterns of cutaneous leiomyosarcoma [30], inflammatory myofibroblastic tumors [32], and malignant fibrous histiocytoma [37] were also identified by the study in single samples.

Another tumor that stood out after consultation was a clear cell hidradenocarcinoma mimicking the immunohistochemistry of a metastatic lobular carcinoma by presenting a positive estrogen receptor, a progesterone receptor, mammaglobin, and CK7 cytokeratin [27]. Finally, the anatomopathological analysis can be highlighted, which identified invasion in different tissues in 19.2% (n = 5) of the samples [24,29,34], with one (1) in the bone marrow [29], one (1) in the right scapular region [24], and three (3) in the axillary lymph node region [19,34,38].

### 3.15. Management

Surgical resection of the tumor in the pacemaker pocket was the most performed procedure among study patients (34.61%, n = 9) (Appendix A) [25,26,30,31,32,37,39,41,42]. Then came the use of chemotherapy (30.76%, n = 8) [33,34,35,38,40,42,44] and radiotherapy (30.76%, n = 8) [24,32,34,38,40,41,43,44].

The use of chemotherapy with cyclophosphamide, doxorubicin, vincristine, and prednisone was described by Rasmussen et al. [34], while Zonca et al. [40]; Bhandarkar, Bewu, and Taylor [33]; and Rothenberger-Janzen, Flueckiger, and Bigler [42] described the use of antiestrogen therapy with tamoxifen. Biran et al. [44] performed chemotherapy with cytoxan, methotrexate and 5-fluoracil. The studies by Moruzzo et al. [35] and Zafirocopoulos and Rouskas [38] also reported the use of chemotherapy but did not present the drugs used.

Radiotherapy was described in 34.61% of cases (*n* = 9) [24,27,32,34,38,40,41,43,44]. Of the nine studies with radiotherapy, only three provided details on how the procedure was performed. The description presented by Hamaker et al. [24] in their study included the use of cobalt therapy with 2000 rads at the pacemaker site, Hojo et al. [43] reported the use of 30 Gy at the tumor site in the pacemaker pocket of a patient with AC, and Khamooshian [41] described the use of 60 Gy of adjuvant radiotherapy in the treatment. However, Reyes [27] presented a patient who refused the recommended irradiation.

Other procedures employed were drainage of hematoma [29], antibiotic therapy for suspected infection at the site of the pacemaker bag after hospital admission [19,28,29], simple mastectomy [38], and unilateral mastectomy with unilateral lymphadenectomy [34,44].

### 3.16. Clinical Outcomes

Regarding the clinical outcomes, seven patients died [24,27,34,35,37,38,40] and nine were cured after surgical intervention, chemotherapy, or radiotherapy [25,26,29,30,31,32,39,42,43]. One patient was described as still undergoing radiotherapy [41], one patient was in palliative care [28], and one patient [19] was undergoing outpatient follow-up with no therapeutic or palliative proposal yet. Four patients did not have the outcomes recorded in their respective studies [19,33,36,44]. These data can be seen in Appendix A.

## 4. Discussion

Every year, approximately 1 million pacemakers are implanted worldwide [46], with a total of 19 million in the USA over the period from 1993 to 2009. Nevertheless, only 15 cases of malignancies are described in the literature, which substantiates the infrequent occurrence of malignancies within the pacemaker pocket [28]. Publications reporting and describing the appearance of tumors with a primary site in the pocket of these devices are still rare in the current medical literature.

The process of formation of malignant neoplasms around the cardiac pacemaker is still not well understood. One hypothesis would be that titanium is involved in tumor formation in this region [28,45]. The wearing away and corrosion of titanium in the human body, associated with the release of metal ions in the reaction, can cause toxicity in the body due to the potential pro-inflammatory effects mediated by interleucins, tumor necrosis factor α (TNF-α), transforming growth factor β (TGF-β), and β-glucuronidase (GLU), and oxidative effects on the tissue surrounding the metal may cause apoptosis, genomic instabilities, and production of titanium-specific T-lymphocytes [45,47,48,49,50]. In addition, titanium particles in their oxidized form (TiO_2_) are capable of inducing DNA damage, genetic mutations, DNA deletions, and formation of micronuclei indicative of chromosomal aberrations in different cell lines [45,51,52].

Another possibility is that there may be genetic factors involved in the neoplasia formation process within the pacemaker pocket. The occurrence of cancer at the site of silicone breast implantation has been linked to mutations in the JAK1/STAT3 signaling pathway and the TP53 gene, which are involved in the modulation and prevention of clonal expansion of tumor cells [53,54,55]. Overexpression of the MYC gene and mutations in BRCA1/2 have also been associated with a higher likelihood of lymphomagenesis in patients following silicone implants [53,56,57]. Mutations in the p53 gene have also been described in the development of oral cancer after metal implants [58,59]. Currently, there is no evidence to support the presence of any specific common genetic component for the development of cancer in the pacemaker pocket among the patients described in the study.

Two other hypotheses regarding the formation of tumors in the pacemaker pocket would be chronic inflammation due to prolonged pacemaker presence and as a result of electrical stimulation [24,28]. The inflammation and the electric stimulation caused by a pacemaker are linked to chronic mechanical irritation, electrochemical disbalance, and trauma, all of which contribute to the development of cancer by inducing prolonged immune activation, cellular damage, and cancer cell migration resulting from electrical activity (galvanotaxis/electrotaxis) [60,61,62].

In the current study, titanium was the main metal described in the composition of pacemakers (30.7%, n = 8) [24,26,28,30,31,43,44]. Moreover, Pinchasov et al. [63] also reported in their study a series of cases of oral cancer of the squamous cell type in patients with dental implants made of titanium. For Onega et al. [64], in their meta-analysis with patients undergoing total arthroplasty who developed cancer at the implant site, there was no identification of an increased risk of cancer with the use of metal prostheses.

A study carried out with the population of Denmark observed a greater chance of developing bladder cancer and multiple myeloma in patients with a cardiac pacemaker, suggesting that the use of the device or the shared risk factors between cancer and cardiovascular diseases would be involved in the development of these tumors [65].

The most frequent comorbidities in the publications were arterial hypertension (20.0%) [28,29,35], DM2 (15.3%) [28,29], and cardiomyopathy (15.3%) [29]. These cardiovascular diseases have several well-established risk factors in common with cancer, such as smoking, alcoholism, physical inactivity, unhealthy diet, dyslipidemia, hypertension, age, obesity, and diabetes. These factors could explain the occurrence of the two pathologies concomitantly [11,66,67,68].

The presence of lymphomas was reported in 23.07% of the observed studies (n = 6) [19,28,29,35,43]. A similar result was described by Kricheldorff et al. [69], who also reported the predominance of lymphoma-type cancer—more specifically, anaplastic large cell lymphoma (BIA-ALCL)—in women undergoing silicone breast implantation. BIA-ALCL is a type of non-Hodgkin’s lymphoma that also originates from a silicone implant in the pectoral region of women [70]. Its genesis involves factors related to immunological interaction between tissue and prosthesis, bacterial growth in the implanted site, and genetic factors [70,71].

The most common clinical manifestation was local expansion over or near the pacemaker pocket, which was reported in terms of either local expansion (*n* = 5) [19,24,28,29,37] or cutaneous nodules (*n* = 11) [26,27,30,31,33,34,35,38,41,42,43,44], in addition to infection (*n* = 1) [34], ulceration (*n* = 5) [29,30,31,34,40], necrosis (*n* = 2) [24,43], fever (*n* = 3) [24,32,35], discharge (*n* = 1) [34], and erythema [19] (*n* = 1). The constant presence of a foreign body, the pacemaker, can lead to chronic inflammation, tissue damage, and production of reactive oxygen species, which can increase the risk of tumor development and the proliferation of microorganisms [72,73].

As for tumor development time, 12 patients developed cancer within 4 years of pacemaker implantation [19,24,27,30,31,34,35,37,38,41,44], 5 patients were diagnosed in a period greater than 4 years and less than 10 years [26,36,38,40,43], and 8 developed cancer over a period of more than 10 years [19,28,29,32,33,39,42]. In comparison, the International Agency for Research on Cancer (IARC) [74] described the growth of sarcomas and lymphomas at sites of orthopedic metal implants as occurring over variable periods, with cases reported between a few months and 30 years after implantation but with an average time of diagnosis that was also less than 10 years.

Chest radiography (26.9%) was the most used imaging test due to its ability to analyze the insertion of the cardiac device and identify tumors or structures adjacent to the implant [75,76], and it was also used in some patients to assess metastases [38,42], an action that is recurrent in underdeveloped countries [77]. Of the patients who underwent computed tomography (26.9%) (n = 7), in four of them the exact location of the mass and its relationship with adjacent structures (pacemaker bag and muscle bundles) were verified, in addition to definition of the presence or absence of lymphadenopathy in the axillary and mediastinal chains [29,32,35,41,78,79,80]. Finally, the third most used imaging test was USG (19.2%) due to its usefulness in observing soft tissues and differentiating the described tumors from isolated abscesses [81] and because it serves as a quick and cost-effective way to locate a lesion in relation to the pacemaker pocket [82], as seen in the studies by De Mattia [36], Khamooshian et al. [41], Zonca et al. [40], and Reyes [27].

In this systematic review, the approaches described were heterogeneous in terms of treatment, with significant differences between them. The treatment was carried out according to the specific indications of each histological type of tumor, the resources and evidence available at the time of the studies’ respective publications and the clinical condition of the patients and the indications for surgery [29,32,35,41,78,79,80], chemotherapy [33,34,35,38,40,42,44], and radiotherapy [24,32,34,38,40,41,43,44].

Primary pacemaker pocket tumors are rare. The relationship between pacemaker components and the appearance of malignancies is not yet well understood, but occurrence is probably due to coincidence [28,83]. Currently, there is insufficient evidence to establish a clear link between the occurrence of pacemaker pocket tumors and a specific factor, such as genetic characteristics, pacemaker composition, or immunological processes. Due to the rarity of these tumors, there is a lack of consensus on the approach to diagnosis and treatment. More studies are needed to improve our understanding of the biology and treatment of these rare tumors.

## 5. Conclusions

Patients who have been implanted with a pacemaker should be routinely clinically evaluated for the occurrence of malignant tumors at the implantation site of these devices.

## Figures and Tables

**Figure 1 cancers-15-05206-f001:**
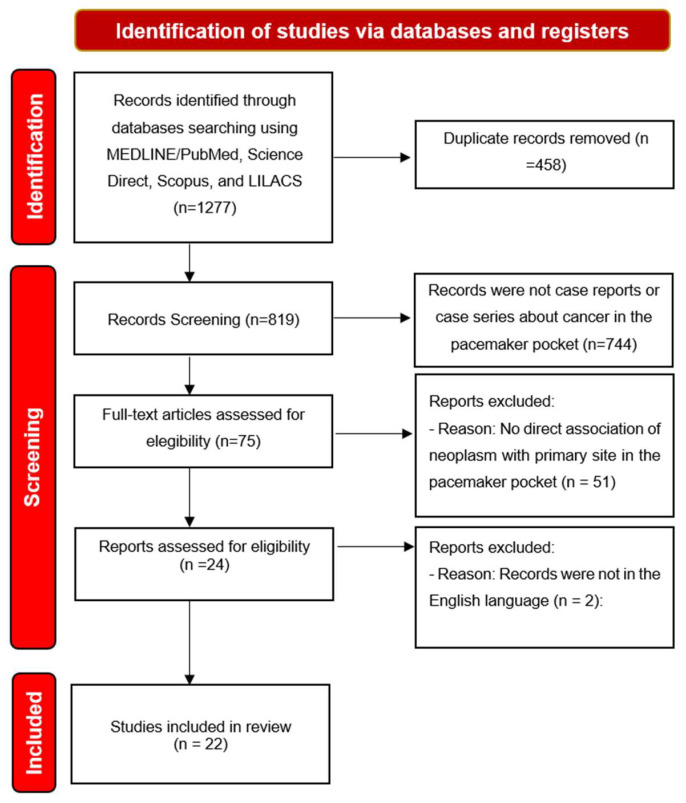
Diagram of the research selection flow adapted from the Preferred Reporting Items for Systematic Reviews and Meta-Analyses (PRISMA).

**Figure 2 cancers-15-05206-f002:**
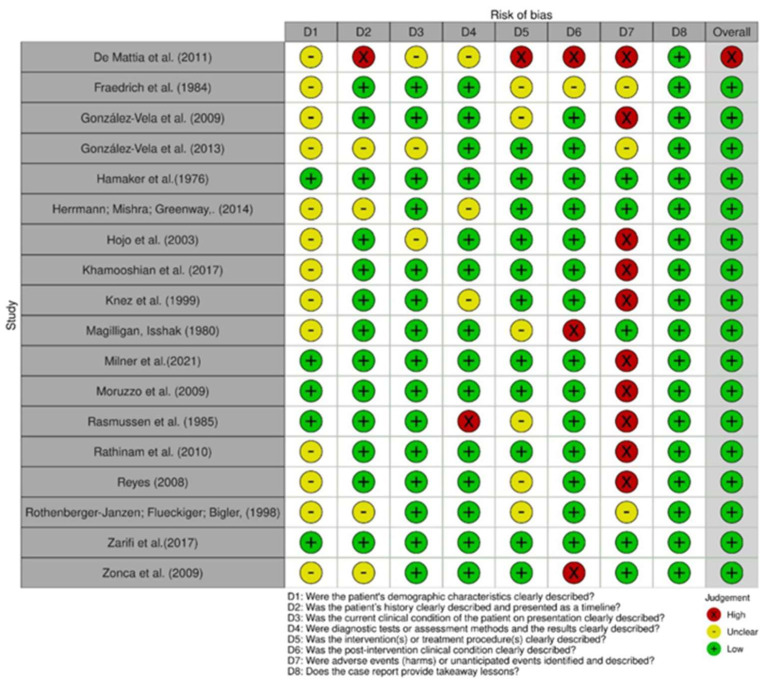
Risk of bias among case reports [24,25,26,27,28,29,30,31,32,34,35,36,37,39,40,41,42,43].

**Figure 3 cancers-15-05206-f003:**
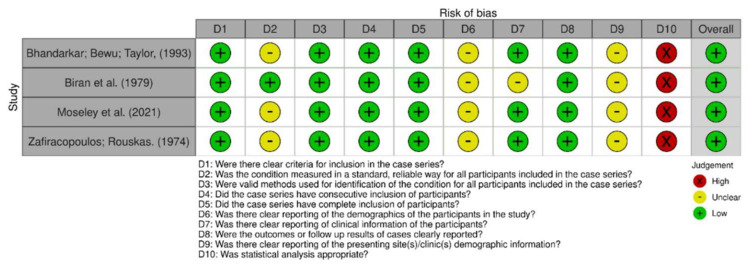
Risk of bias among case series [19,33,38,44].

**Figure 4 cancers-15-05206-f004:**
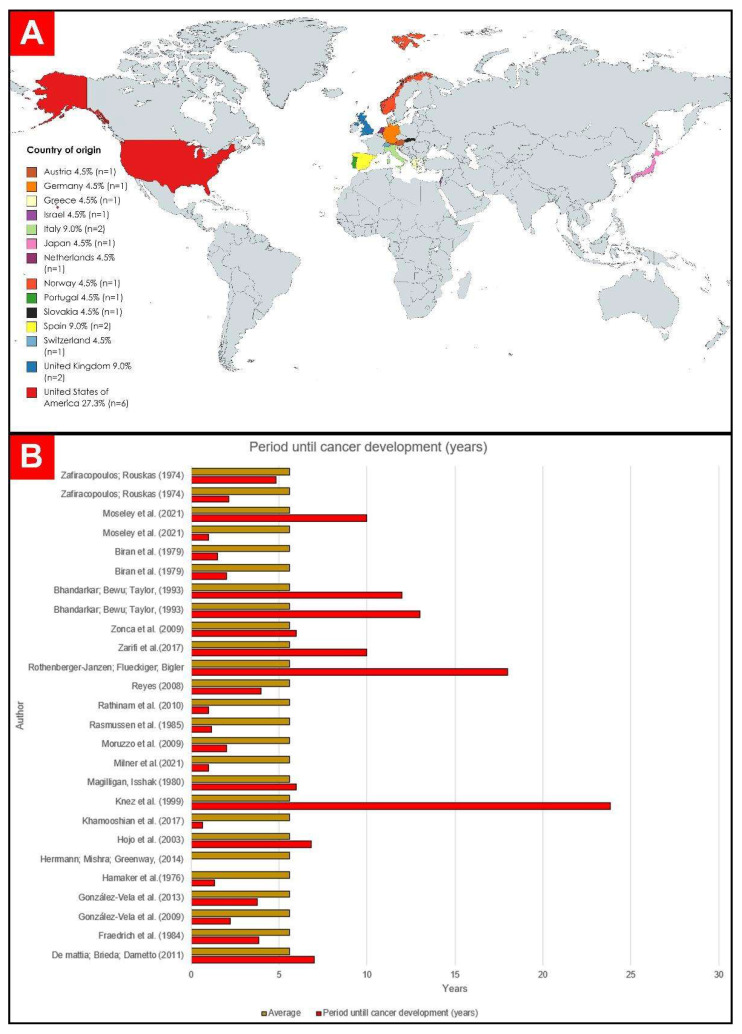
(**A**) Countries of origin of the selected publications. (**B**) Graphic representation of time to development of cancer in years [19,24,25,26,27,28,29,30,31,32,33,34,35,36,37,38,39,40,41,42,43,44].

**Table 1 cancers-15-05206-t001:** Identification of articles included in the study.

Author	Journal	Year of Publication	Country of Origin	Study Type
Bhandarkar, Bewu, and Taylor [33]	*Postgrad. Med. J.*	1993	United Kingdom	Case series
Biran et al. [44]	*Journal Surg. Oncol.*	1979	Israel	Case series
De Mattia, Brieda, and Dametto [36]	*Europace*	2011	Italy	Case report
Fraedrich et al. [37]	*Thorac. Cardiovasc. Surg.*	1984	Germany	Case report
Gonzalez-Vela et al. [30]	*Dermatol. Surg.*	2009	Spain	Case report
González-Vela et al. [31]	*Cardiovasc. Pathol.*	2013	Spain	Case report
Hamaker et al. [24]	*Ann. Thorac. Surg.*	1976	United States	Case report
Herrmann, Mishra, and Greenway [25]	*Dermatol. Surg.*	2014	United States	Case report
Hojo et al. [43]	*Int. J. Hematol.*	2003	Japan	Case report
Khamooshian et al. [41]	*HeartRhythm Case Rep.*	2017	Netherlands	Case report
Knez et al. [39]	*Pacin. Clin. Electrophysiol.*	1999	Austria	Case report
Magilligan and Isshak [26]	*Pacin. Clin. Electrophysiol.*	1980	United States	Case report
Milner et al. [29]	*J. Cardiol. Cases*	2021	Portugal	Case report
Moruzzo et al. [35]	*Leuk. Lymphoma*	2009	Italy	Case report
Moseley et al. [19]	*HeartRhythm Case Rep.*	2021	United States	Case series
Rasmussen et al. [34]	*Pacin. Clin. Electrophysiol.*	1985	Norway	Case report
Rathinam et al. [32]	*Interact. Cardiovasc. Thorac. Surg.*	2010	United Kingdom	Case report
Reyes [27]	*Pacin. Clin. Electrophysiol.*	2008	United States	Case report
Rothenberger-Janzen, Flueckiger, and Bigler [42]	*Pacin. Clin. Electrophysiol.*	1998	Switzerland	Case report
Zafiracopoulos and Rouskas [38]	*Lancet*	1974	Greece	Case series
Zarifi et al. [28]	*J. Cardiol. Cases*	2017	United States	Case report
Zonca et al. [40]	*Bratisl. Lek. Listy.*	2009	Slovakia	Case report

**Table 2 cancers-15-05206-t002:** Characteristics of patients and pacemakers.

Author	Age	Sex	PM Model	PM Construct	Type of Cancer	Clinical Findings	Comorbidities	Period of Cancer Onset
**Case Reports**								
De Mattia, Brieda, and Dametto (2011) [36]	87	Woman	NA	NA	Invasive ductal carcinoma	Firm, coin-sized lesion with skinerosion	NA	7 years (84 months)
Fraedrich et al. (1984) [37]	82	Man	First: Medtronic Xytron (1980) Second: Cordis Stanicor (1981)	NA	Malignant fibrous histiocytoma	Severe swelling in the PP area	AV block, Stokes-Adams syndrome	3 years and 10 months (46 months)
González-Vela et al. (2009) [30]	74	Man	First: Intermedics 254-31 cardiac pacemaker (1992), second: Guidant Meridian SSI 116-120 (2002)	First: Titanium Second: ND	Cutaneous leiomyosarcoma	Ulcerated nodular skin lesion in the subpectoral PP	SNS	2 years and 3 months (27 months)
González-Vela et al. (2013) [31]	89	Man	Medtronic KVDD901 Kappa 900 VDD	Titanium	Atypical fibroxanthoma	Ulcerated skin nodule beneath PP	AV block	3 years and 9 months (45 months)
Hamaker et al. (1976) [24]	48	Man	Medtronic 5841	Titanium coated	Plasmacytoma	Necrosis of the skin on PM, edema, fatigue, fever, pallor, cardiac murmurs	Valvulopathy, AV block	1 year and 4 months (16 months)
Herrmann, Mishra, and Greenway (2014) [25]	75	Man	NA	NA	Nodular basal cell carcinoma	Erythematous plaque on the left pectoral over PM	AF with slow ventricular response	NA
Hojo et al. (2003) [43]	29	Man	NA	Titanium	Diffuse large B-cell lymphoma, stage II	Palpable, painless mass in PP; necrosis; keloid scar; lymphadenopathy in the left armpit	AV block, Stokes-Adams syndrome	6 years and 10 months (82 months)
Khamooshian et al. (2017) [41]	43	Man	NA	NA	Undifferentiated pleomorphic sarcoma	Large, firm, palpable mass cranial to the PP	Intermittent AV block of unknown origin	8 months
Knez et al. (1999) [39]	84	Man	Medtronic CapSure SP 4024	NA	Invasive ductal carcinoma	PP ulcer in the upper quadrant of the right pectoral	NA	23 years and 10 months (286 months)
Magilligan and Isshak (1980) [26]	89	Woman	Medtronic 5950	Titanium	Breast adenocarcinoma	Palpable mass at PP	NA	6 years (72 months)
Milner et al. (2021) [29]	78	Man	VDDR Pacemaker (BostonScientific INSIGNIA™ I AVT VDR 882)	NA	lymphoplasmacytic lymphoma	Local discomfort, bag expansion, ulcerative lesion	Non-ischemic cardiomyopathy; HF, AV block, SAH, DYS, T2DM, CKD; paroxysmal AF; vocal cord cancer	12 years (144 months)
Moruzzo et al. (2009) [35]	68	Man	NA	NA	Stage IE non-Hodgkin’s lymphoma	Fever, palpable mass in the region of the PP	Ex-smoker, SAH, TIA, syncope	2 years (24 months)
Rasmussen et al. (1985) [34]	75	Man	Unipolar Cordis Stanicor (Cordis Corp, Miami, FL, USA)	Mercury zinc	Papillary adenocarcinoma	Palpable mass over battery site, inflammation with ulceration and purulent discharge	AF, AV block, HF	1 year and 2 months (14 months)
Rathinam et al. (2010) [32]	64	Man	NA	NA	Inflammatory myofibroblastic tumor	Cough, high fever, night sweats, lethargy, loss of appetite, weight loss, and palpable mass	Stokes-Adams syndrome	First surgery: 10 years (120 months); second surgery: 1 year (12 months)
Reyes (2008) [27]	88	Woman	NA	NA	Clear cell hidradenocarcinoma	Palpable, progressively enlarging, painless mass in the right hemithorax	NA	4 years (48 months)
Rothenberger-Janzen, Flueckiger, and Bigler (1998) [42]	90	Woman	Leptos VVI 01-A	NA	Intraductal adenocarcinoma with extracellular mucus	Palpable, irregular, painless mass in the lateral quadrant of the left breast over the site of PM	NA	18 years (216 months)
Zarifi et al. (2017) [28]	100	Man	Medtronic Adapt ADDR01	Titanium	Plasmablastic lymphoma	PP expansion, chest pain	T2DM, HT, SAH, SNS	10 years (120 months)
Zonca et al. (2009) [40]	78	Woman	NA	NA	Invasive ductal carcinoma	Ulcer at the PM site; inflammatory	NA	6 years (72 months)
**Case Series**								
Bhandarkar, Bewu, and Taylor (1993) [33]	83	Woman	First: Cordis Omni Pacemaker (1980) Second: Cordis 337A VVI type (1986)	NA	Adenocarcinoma	Palpable mass in the lower lateral quadrant of the left breast	Sinus bradycardia	13 years (156 months)
	84	Woman	First: Teletronics, VVI model 120B (1981) Second: Optima MP 580 VVI (1991)	NA	Adenocarcinoma	Palpable mass located directly over the site of the second PM	SNS	12 years (144 months)
Biran et al. (1979) [44]	66	Woman	Medtronic 5942 pacemaker	Titanium	Intraductal carcinoma	Palpable mass in the right breast just above the PM site	Carcinoma of the left breast, fibrotic changes in the left lung, AV block	2 years (24 months)
	65	Woman	Medtronic 5942 pacemaker	Titanium	Adenocarcinoma and Paget’s disease	Papillary discharge on right breast, mass above the right nipple	AV block	1 year and 6 months (18 months)
Moseley et al. (2021) [19]	90	Woman	NA	NA	Large B-cell lymphoma	Swelling, tenderness, and erythema of the PP on the left chest	AV block, CKD, SAH, CVA, RA, T2DM	1 year (12 months)
	84	Man	NA	NA	B-cell lymphoma and moderately differentiated squamous cell carcinoma	Cyst located in the left infraclavicular region, warm and sensitive to palpation	AV block, AF, T2DM, SAH	10 years (120 months)
Zafiracopoulos and Rouskas (1974) [38]	63	Woman	Vitatron Demand Pacemaker	NA	Schirroous adenocarcinoma	Lump near the PM implant, palpable	AV block and Stokes-Adams syndrome	2 years and 2 months (26 months)
	68	Woman	Vitatron Demand PM in the right breast (1969), replaced in the left breast (1970)	NA	Schirroous adenocarcinoma	Painless and irregular mass, fixed to the skin	AV block and Stokes-Adams syndrome	4 years and 1 month (49 months)

Abbreviations: NA: not available; AV: atrioventricular; AF: atrial fibrillation; HF: heart failure; SAH: systemic arterial hypertension; T2DM: type 2 diabetes mellitus; CKD: chronic kidney disease; CVA: cerebrovascular accident; RA: rheumatoid arthritis; PP: pacemaker pocket; PM: pacemaker; SNS: sinus node syndrome; HT: hypothyroidism; TIA: transient ischemic attack; DYS: dyslipidemia.

## Data Availability

Data sharing is not applicable to this article.

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
