# Peer review of "Malignant Neoplasms Arising in the Cardiac Pacemaker Cavity: A Systematic Review"

_cancers, 2023, doi:10.3390/cancers15215206_

Round 1

Reviewer 1 Report

I will like to congratulate the authors for their effort. The overall study is well presented with attention to details. The structure of the manuscript is correct.

Reviewer 2 Report

In this systematic review the authors explored the literature of cardiac pacemaker pocket tumors. Primary tumors of the heart are very rare, however an association between implantable pacemakers and neoplasia around the pacemaker tissue has been postulated and recorded.

Concerns
1. The syntax of the scientific question of this systematic review should be revised, the current version: "How  are  the  clinical characteristics and management used in patients with neoplasms in pacemaker pocket?", is not clear.
2. The search strategy section (2.3) strongly suggests that the aim of this systematic review is the association between implantable electronic devices and pacemaker pocket tumors. The authors should re-phrase their question and clarify their aims in the text.
3. If the authors would like to compare the incidence of sporadic pacemaker pocket tumors to pacemaker pocket tumors due to implantable electronic device, this should be also clearly stated.
4. The authors consider only articles with confirmed cancerous histopathologies. What about benign pacemaker pocket tumors? These could be also detrimental for the patients even if not cancerous. Should the authors consider to include these reports separately to investigate the risk of implantable electronic devices in heart tissue histopathologic changes?
5. What is the incidence of development pacemaker pocket tumors when compared to the number of implantation of cardiac electronic devices?

Reviewer 3 Report

1. The article is interesting in itself, but the claims made are questionable.

2. There is no clear causal relationship between pacemaker implantation and the development of malignancy. It is not here and malignancy can develop anywhere.

3. Literary sources are old enough.

4. Unclear genetic predisposition of subjects to the oncological process.

5. Claims regarding titanium, and chronic inflammation are at best based on 1-2 literature sources and are more speculative.

6. There is no confirmation or verification that the stimulator lodge (pocket) is the primary oncological focus.

1. The article is interesting in itself, but the claims made are questionable.

2. There is no clear causal relationship between pacemaker implantation and the development of malignancy. It is not here and malignancy can develop anywhere.

3. Literary sources are old enough.

4. Unclear genetic predisposition of subjects to the oncological process.

5. Claims regarding titanium, and chronic inflammation are at best based on 1-2 literature sources and are more speculative.

6. There is no confirmation or verification that the stimulator lodge (pocket) is the primary oncological focus.

Round 2

Reviewer 2 Report

There is an increase number of case reports of pacemaker pocket tumors associated with implantable cardiac electronic devices. This is an interesting systematic review but the authors should re-phrase their scientific question and to consider the inclusion of case report of benign pacemaker pocket tumors as well. The authors responded well to reviewer's concerns.

Author Response

Once again, we would like to express our gratitude for your contribution in reviewing the previously made changes. We have revised the guiding scientific question in order to better align it with the text, changing it from "Are arising malignancies on pacemaker pockets correlated to the implanted cardiac devices and what are the clinical/demographic characteristics of patients that bolster this hypothesis?" to "What are the main clinical, demographic, and management-related characteristics described in cases of malignancy affecting the region covering the pacemaker implantation cavity?", as can be reviewed in the third paragraph of section "2.2. Eligibility Criterion," page 3 of 19.

Regarding the recommendation to add benign tumors, we agree that it would indeed be of great interest to include them in the paper. However, when we registered the study in PROSPERO under the number CRD42022360240, we delimited our analysis to cases describing only malignant tumors in the region. The addition of benign tumors could potentially lead to discrepancies with the submitted protocol.